# Effect of DHA-Enriched Phospholipids from Fish Roe on Rat Fecal Metabolites: Untargeted Metabolomic Analysis

**DOI:** 10.3390/foods12081687

**Published:** 2023-04-18

**Authors:** Xiaodan Lu, Luyao Huang, Yanjun Chen, Ling Hu, Rongbin Zhong, Lijiao Chen, Wenjian Cheng, Baodong Zheng, Peng Liang

**Affiliations:** 1Engineering Research Centre of Fujian-Taiwan Special Marine Food Processing and Nutrition Ministry of Education, Fuzhou 350002, China; 2College of Food Science, Fujian Agriculture and Forestry University, Fuzhou 350002, China

**Keywords:** DHA-enriched phospholipids, lipid metabolism, differential metabolites, pathways

## Abstract

Lipid metabolism disorder has become an important hidden danger threatening human health, and various supplements to treat lipid metabolism disorder have been studied. Our previous studies have shown that DHA-enriched phospholipids from large yellow croaker (*Larimichthys Crocea*) roe (LYCRPLs) have lipid-regulating effects. To better explain the effect of LYCRPLs on lipid regulation in rats, the fecal metabolites of rats were analyzed from the level of metabolomics in this study, and GC/MS metabolomics measurements were performed to figure out the effect of LYCRPLs on fecal metabolites in rats. Compared with the control (K) group, 101 metabolites were identified in the model (M) group. There were 54, 47, and 57 metabolites in the low-dose (GA), medium-dose (GB), and high-dose (GC) groups that were significantly different from that of group M, respectively. Eighteen potential biomarkers closely related to lipid metabolism were screened after intervention with different doses of LYCRPLs on rats, which were classified into several metabolic pathways in rats, including pyrimidine metabolism, the citric acid cycle (TCA cycle), the metabolism of L-cysteine, carnitine synthesis, pantothenate and CoA biosynthesis, glycolysis, and bile secretion. L-cysteine was speculated to be a useful biomarker of LYCRPLs acting on rat fecal metabolites. Our findings indicated that LYCRPLs may regulate lipid metabolism disorders in SD rats by activating these metabolic pathways.

## 1. Introduction

Metabolomics, an essential branch of systems biology, focuses on the biological pathways related to disease pathogenesis and reveals the biological effects of treatment by exploring the variations of endogenous metabolites [1,2,3]. Metabolite comparisons and data analyses are two important parts of metabolomics research. A data analysis is performed to find out the overall differences among target samples, including in data collection and processing and dimensional reduction processing [4]. Metabolites can be comparatively analyzed by employing techniques such as mass spectrometry, chromatography, and spectroscopy [5]. At present, metabolomics has been successfully used in medical fields, such as for hemolytic anemia [6], type 2 diabetes [7,8,9], Parkinson’s disease [10], and hypertension [11]. Additionally, metabolomics plays a vital role in studying the mechanisms of natural active substances relating to organisms. Its wide application in the food industry [12] implies that metabolomics is currently one of the most promising techniques for identifying the mechanisms of functional substances.

Lipid metabolism disorders are the key factors in the development of metabolic syndrome [13], which is characterized by dyslipidemia, insulin resistance, abdominal obesity, and high blood pressure [14]. The excessive intake of fat has been shown to cause dysfunction and diseases of many organs, such as fat tissue metabolism, liver fat storage, pancreatic islet function, and others [15,16,17]. High blood lipid levels, liver lipid accumulation [18], obesity [19], non-alcoholic fatty liver disease (NAFLD) [20], diabetes [21], and atherosclerosis [22] are mainly caused by lipid metabolism disorder. Disorders of lipid metabolism are often closely related to factors such as having a long-term irregular diet [23,24]. Thus, how to adjust the lipid metabolism disorder by changing the diet structure has become a research hotspot. Additionally, the mechanism regulating the metabolic disorder is complex and cannot currently be fully explained by a single mechanism.

Marine animals are rich in n-3 polyunsaturated fatty acids (n-3 PUFAs) [25,26], and insufficient intake of n-3 PUFAs is one of the dietary factors that are harmful to health [27]. The n-3 PUFAs from marine foods often exist in the form of phospholipids (PLs), such as docosahexaenoic acid PLs (DHA-PLs) and eicosapentaenoic acid PLs (EPA-PLs). Marine DHA/EPA-PLs have received more and more attention due to their beneficial health effects, which have been reported to significantly regulate dyslipidemia [28] and prevent cardiovascular diseases [29], NAFLD [30,31], atherosclerosis [32], hypertension [33], allergies [34], and neurodegenerative pathologies [35]. It is well-known that fish roes [36] are widely consumed by humans due to their high contents of DHA/EPA-PLs [37,38,39]. In China, large yellow croaker (*Larimichthys Crocea*) roe is a kind of major byproduct containing a high amount of DHA-PLs [40]. LYCRPLs have great nutritional value [41], and their inhibitory effect on the accumulation of triglycerides was studied at the cellular level [42] in our early research. Previous animal experiments have also shown that LYCRPLs can significantly regulate lipid metabolism and improve the intestine microbiota disorder induced by a high-fat diet [43]. Nevertheless, there is no report about the effect of LYCRPLs on the fecal metabolites of rats with a high-fat diet.

Therefore, GC/MS non-targeted metabolomics measurements were used to determine and analyze the differential metabolites and metabolic pathways related to lipid metabolism in the feces of SD rats after LYCRPL treatment. To better explain the effect of LYCRPLs on lipid regulation in rats, the fecal metabolites of rats were analyzed from the perspective of metabolomics in this study.

## 2. Materials and Methods

### 2.1. Materials

The roe of *Larimichthys Crocea* was obtained from Fujian Yuehai Aquatic Food Co., Ltd. (Ningde, China). All chemicals and solvents were either of analytical or HPLC grade. The water, methanol, pyridine, n-hexane, methoxylamine hydrochloride (97%), and BSTFA with 1% TMCS were bought from CNW Technologies GmbH (Düsseldorf, Germany). The trichloromethane was sourced from Sinopharm Chemical Reagent Co., Ltd. (Shanghai, China). The L-2-chlorophenylalanine came from Shanghai Hengchuang Bio-technology Co., Ltd. (Shanghai, China).

### 2.2. Preparation of LYCRPLs

The LYCRPLs were obtained in the laboratory by referring to the methods used in our previous study [43]. The thawed *Larimichthys Crocea* roe was cut into small pieces and freeze-dried. The neutral fat was removed using the supercritical CO_2_ fluid method, and 95% ethanol was added at a ratio of 1:10. The extraction was performed at 40 °C for 30 min in three repetitions. The extract was recovered via filtration and freeze-dried to obtain LYCRPLs, which were stored at −18 °C for future use.

### 2.3. Animal Treatment and Sample Collection

After the experimental Sprague–Dawley (SD) rats adapted to the environment, 50 male SD rats (specific-pathogen-free, SPF, 120 ± 10 g) were randomly divided into five groups: the control (K) group, the model (M) group, the low-dose (GA) group (0.005 g·mL^−1^), the medium-dose (0.015 g·mL^−1^) group (GB), and the high-dose (0.03 g·mL^−1^) (GC) group. The experiment was carried out under the same conditions as our previous study [43]. The rats were kept in a comfortable environment to the utmost extent throughout the experiment. The samples were given by intragastric administration (dose: 0.01 mL·g^−1^) at a fixed time every day for 8 weeks. The nutritional information for the diet used is shown in Table 1. SD rat feces samples were collected one day prior to the animal dissection experiment, sealed with a parafilm in a sterilized EP tube, and stored in a refrigerator at −80 °C for further processing.

### 2.4. Sample Processing

A 60 mg feces sample and 40 μL of internal standard (0.3 mg/mL 2-chloro-l-phenylalanine in methanol) plus 360 μL of extraction solvent (methanol/water, 4/1 *v*/*v*) were transferred to a 1.5 mL Eppendorf tube containing two steel balls. The samples were stored at −20 °C for 2 min, then ground at 60 Hz for 2 min. Next, 200 μL of chloroform was added and the samples were ultrasonically extracted for 30 min at room temperature. After 30 min, the samples were stored at −20 °C for another 30 min, then centrifuged at 13,000 rpm for 10 min at 4 °C. A Quality control (QC) sample was formed by combining aliquots of all samples, and 300 μL of the supernatant was transferred to a glass vial, followed by vacuum-drying at room temperature. Next, 80 μL of methoxylamine hydrochloride (dissolved in pyridine, 15 mg/mL) was added, vortexed for 2 min, and incubated at 37 °C for 90 min. Then, 80 μL of BSTFA (with 1% TMCS) and 20 μL n-hexane were added, vortexed for 2 min, and derivatized at 70 °C for 60 min. The samples were left at room temperature for 30 min before their analysis by GC-MS.

### 2.5. Gas Chromatography–Mass Spectrometry Analysis

An Agilent 7890B GC-5977A MSD system (Agilent Tech., Santa Clara, CA, USA) was used to analyze the samples with a HP-5MS fused silica capillary column (30 m × 0.25 mm × 0.25 μm). Helium (99.999%) was used as the carrier gas (1 mL/min) and the injector temperature was 260 °C, with a 5 min solvent delay, being performed in splitless mode. The oven temperature was initially set to 60 °C, then ramped to 125 °C (8 °C/min), 210 °C (5 °C/min), 270 °C (10 °C/min), and 305 °C (20 °C/min) and held at 305 °C for 5 min. The MS quadrupole and ion source temperatures (electron impact) were 150 °C and 230 °C, respectively, with a collision energy of 70 eV; the data were collected in full-scan mode (50–500 *m*/*z*). Quality control was performed regularly for repeatability.

### 2.6. Data Processing

The Analysis Base File Converter software was used to convert the raw data (.D format) to .abf, which was then imported into MD-DIAL for data processing. The LUG database was utilized to annotate metabolites. The “statistic compare” component was used to generate the raw “data array” (.txt) with three-dimensional data sets (Appendix A), including the sample information, peak names and retention times (*m*/*z*), and peak intensities. Internal standards and any pseudo-positive peaks were removed. RSDs >0.3 for the interior label were deleted, and the peak strength (peak area) was normalized by the retention time partition period. Log10 transformation was applied to the data, which were then imported into the R ropls package.

### 2.7. Statistical and Pathway Analysis

The experimental data are presented as the means ± SDs. To determine the differences among groups, a one-way ANOVA followed by Duncan’s multiple range test was conducted using SPSS (version 13.0). A *p*-value of <0.05 indicated statistical significance, and <0.01 was considered extremely significant. Figure 3A was generated using Origin 9.0 software. The differential metabolites of each group were analyzed after comparison with the KEGG, HMDB, and LipidMaps databases.

## 3. Results

### 3.1. The Effect of LYCRPLs on the Metabolites of Rats with a High-Fat Diet

QC samples are used to equilibrate the system prior to sample detection and to assess the stability of the mass spectrometry system during sample detection. A total ion chromatographic (TIC) overlap comparison of the QC samples is shown in Figure 1A. The results show that the response intensity and retention time of the QC samples remained stable within 48 h, and the chromatographic peaks basically overlapped, indicating that the error caused by the instrument during the experiment was small and the obtained results were highly reliable.

PLS-DA is often used to visually display the differences between groups, and the greater the separation of the two groups of samples in the figure, the more significant the difference. PLS-DA was used to determine differences in metabolites among groups. As can be seen from Figure 1B, the significant separation of each group from group M was observed. The fecal metabolites between group K and group M, group GA and group M, group GB and group M, and group GC and group M were completely separated. Different doses of LYCRPLs can effectively make the predicted main components of rat fecal metabolites migrate in the positive direction of the X axis. The higher the dose, the greater the migration to the positive X axis. The results showed that the fecal metabolites of SD rats with a high-fat diet were significantly different among groups.

Hierarchical clustering of the metabolite expression was performed to more intuitively show the differences in metabolite expression among groups. From the visualization of metabolite differences in Figure 2, it can be seen that the hierarchical clustering boundary of group K and group M is clear (Figure 2A), indicating that a high-fat diet can significantly regulate the expression of fecal metabolites in rats. Group GA (Figure 2B), group GB (Figure 2C), and group GC (Figure 2D) can be clearly distinguished from group M, indicating that LYCRPLs have a significant effect on the fecal metabolites of SD rats.

### 3.2. Screening of Differential Metabolites in the Intestine of Rats with a High-Fat Diet after Intervention by LYCRPLs

The variable importance in projection (VIP) is the weight value of the variable. The larger the VIP value in the PLS-DA model, the greater the contribution of the variable to the grouping. VIP > 1 and *p* < 0.05 are the common criteria for screening differential metabolites between groups. According to the VIP value obtained from the PLS-DA analysis, the differences between groups were further analyzed. It is interesting to note that significant changes in the fecal metabolites were observed after supplementation with different doses of LYCRPLs. The number of different metabolites between group K and group M is as high as 101 (Figure 3A), with 40 significantly being up-regulated and 61 significantly down-regulated (Figure 3B), indicating that a high-fat diet can significantly change the fecal metabolites of experimental rats. Compared with group M, the numbers of differential metabolites in group GA, group GB, and group GC equaled 54 (39 up-regulated and 15 down-regulated), 47 (44 up-regulated and 3 down-regulated), and 57 (53 up-regulated and 4 down-regulated), respectively.

A total of 18 potential biomarkers related to hyperlipidemia were screened in the key metabolic pathways of the intestine (Table 2). Compared with group K, the contents of palmitic acid, ethanolamine, and N-acetyl-d-glucosamine in the intestine of group M were significantly (*p* < 0.05) increased, while the levels of 8 other metabolites were significantly (*p* < 0.05) decreased, including uracil, caproic acid, pyruvic acid, L-alanine, 2-hydroxybutyric acid, and oxoglutaric acid. The metabolites in the intestine were also significantly (*p* < 0.05) changed after supplementation with different doses of LYCRPLs. Compared with group M, the contents of glycolic acid, L-cysteine, and glycerol 3-phosphate in the intestine in group GA were significantly (*p* < 0.05) increased, while the levels of 3-methyl-2-oxovaleric acid and phosphoenolpyruvic acid were significantly (*p* < 0.05) decreased. The contents of L-cysteine, L-glutamine, and pantothenic acid in the intestine in group GB were significantly (*p* < 0.05) increased. The levels of L-cysteine, D-glucose, pantothenic acid, L-lysine, and glycerol 3-phosphate were significantly (*p* < 0.05) increased in group GC, while the levels of 3-methyl-2-oxovaleric acid and oxoglutaric acid were significantly (*p* < 0.05) decreased.

Thus, our results indicated the changes induced by a high-fat diet in the levels of palmitic acid, ethanolamine, N-acetyl-d-glucosamine, uracil, caproic acid, pyruvic acid, L-alanine, 2-hydroxybutyric acid, and oxoglutaric acid, while treatments with different doses of LYCRPLs were associated with the changes in the levels of compounds such as glycolic acid, L-cysteine, glycerol 3-phosphate, 3-methyl-2-oxovaleric acid, phosphoenolpyruvic acid, L-glutamine, pantothenic acid, D-glucose, L-lysine, and oxoglutaric acid.

### 3.3. Metabolic Pathway Enrichment and Analysis of Potential Biomarkers

KEGG is a comprehensive database used to understand the functions and interactions of metabolites in biological systems, which can provide information about metabolic pathways, diseases, and other related metabolites [44]. The most important signal transduction and metabolic pathways involved in the differential metabolites can be determined through KEGG pathway enrichment [45]. In this study, the KEGG database was used to enrich and analyze the metabolic pathways of differential metabolites, which will help to further understand the effect of LYCRPLs on the fecal metabolites of rats with a high-fat diet.

Figure 4 reveals the first 10 pathways among groups. When compared with group K, the KEGG pathways with significant differences in group M (Figure 4A) included the biosynthesis of amino acids, central carbon metabolism in cancer, the GABAergic synapse, lysine degradation, pyrimidine metabolism, citrate cycle (TCA cycle), basal cell carcinoma, phenylalanine metabolism, tyrosine metabolism, and the glucagon signaling pathway. Compared with group M, the KEGG pathways with significant differences in group GA (Figure 4B) included the cAMP signaling pathway, pyrimidine metabolism, morphine addiction, alcoholism, choline metabolism in cancer, the gap junction, the synaptic vesicle cycle, the cGMP-PKG signaling pathway, the regulation of lipolysis in adipocytes, and Parkinson’ disease. The significantly different KEGG pathways in group GB (Figure 4C) included purine metabolism, the cAMP signaling pathway, arginine and proline metabolism, morphine addiction, alcoholism, the gap junction, bile secretion, the synaptic vesicle cycle, the cGMP-PKG signaling pathway, and the regulation of lipolysis in adipocytes. The KEGG pathways with significant differences between group M and group GC (Figure 4D) included bile secretion, the HIF-1 signaling pathway, the glucagon signaling pathway, taste transduction, ABC transporters, central carbon metabolism in cancer, non-alcoholic fatty liver disease (NAFLD), galactose metabolism, the insulin signaling pathway, and the FoxO signaling pathway.

After the comparative analysis with the KEGG, HMDB, and LipidMaps databases and key metabolic pathways (Table 2) of potential biomarkers associated with hyperlipidemia, we found that these substances induced by a high-fat diet were mainly concentrated in pyrimidine metabolism and the citric acid (TCA) cycle metabolic pathway. The metabolites of the LYCRPLs groups are significantly different, and the significant enrichment pathways included the metabolism of L-cysteine, carnitine synthesis, pantothenate and CoA biosynthesis, glycolysis, and bile secretion.

## 4. Discussion

The importance of lipids in the human body is undeniable, since they are related to material transportation, energy metabolism, and metabolic regulation [46,47]. Lipids, especially n-3 PUFAs, have been found to have a significant impact on the intestinal microbiota [48]. They have been associated with changes in bacterial populations, increased microbial diversity, and the promotion of beneficial bacteria. Furthermore, n-3 PUFAs have been shown to reduce oxidative stress [49] and inflammation [50] in the intestine, which can help to protect against diseases such as irritable bowel syndrome and inflammatory bowel disease. Our previous studies have indicated that LYCRPLs could significantly reduce the content of triglycerides in HepG2 cells induced by oleic acid [42]. LYCRPLs have also been shown to regulate lipid metabolism disorders and intestine microbiota imbalances in SD rats with a high-fat diet [43]. In this study, we further investigated the changes of fecal metabolites in rats after supplementation of LYCRPLs. GC-MS metabolomics, along with a multivariate analysis, was employed to detect potential biomarkers. The metabolic pathways and metabolic mechanism after the administration of LYCRPLs are illustrated in Figure 5.

The citric acid (TCA) cycle is a critical part of energy metabolism, which provides the energy necessary for bodily activities. ATP acts as an energy source in the metabolic process and plays a vital role in it [51]. An abnormal TCA cycle leads to metabolic diseases, such as hyperlipidemia and diabetes [52]. Uracil is involved in many enzymatic reactions in the human body, mainly by combining with ribose and phosphate to assist in the synthesis of many enzymes necessary for cell function. The biosynthesis of polysaccharides and the transport of aldoses are also involved. In this study, when the experimental SD rats were fed with a high-fat diet, the pyrimidine metabolism and TCA cycle pathways of the rat metabolites were abnormally reduced, showing down-regulation of uracil and oxoglutaric acid (Figure 6A,B). It can be speculated that abnormalities of the pyrimidine metabolism and TCA cycle may cause hyperlipidemia in SD rats. However, supplementation with high doses of LYCRPLs can significantly increase the content of oxoglutaric acid in fecal metabolites and activate the TCA cycle metabolic pathway in high-fat-diet rats (Figure 6B,C). The results showed that LYCRPLs may participate in the TCA cycle pathway of the intestinal metabolism by up-regulating the expression of oxoglutaric acid, thereby regulating lipid metabolism disorders in SD rats.

Glycolysis is a metabolic pathway composed of ten intermediate compounds, which can convert D-glucose to pyruvate. After pyruvate is produced, it will further undergo pyruvate metabolism, tyrosine metabolism, and pantothenate and CoA biosynthesis. The whole process of glycolysis can be divided into two phases: the chemical initiation phase with D-glucose as the starting compound, and the energy production phase [53,54,55]. Pantothenate and CoA biosynthesis [56] was also identified as a major metabolic pathway. Disorders in these pathways may result in disruption of the energy supply. In our study, a high-fat diet significantly (*p* < 0.01) decreased the metabolic level of pyruvate acid, while low-dose LYCRPL supplementation significantly (*p* < 0.05) increased the expression of glycerol 3-phosphate, which is a chemical intermediate in the glycolysis pathway. Medium-dose LYCRPLs can significantly (*p* < 0.05) increase the expression of pantothenic acid and activate the pantothenate and CoA biosynthesis pathway. The levels of pantothenic acid and D-glucose were significantly (*p* < 0.05) increased in the group treated with high-dose LYCRPLs (Table 2), which activated pantothenate and CoA biosynthesis and enhanced the glycolysis pathway. It is speculated that LYCRPLs may activate the glycolysis pathway by up-regulating the levels of glycerol 3-phosphate, pantothenic acid, and D-glucose (Figure 6D), playing a role in regulating intestine metabolism disorders in rats.

Bile acids (BAs) are the end products of cholesterol breakdown and have an essential function in lipid metabolism [57]. BAs can also improve metabolic capacity by promoting forward intestinal motility [58]. The levels of D-glucose were significantly (*p* < 0.05) increased in the group treated with high-dose LYCRPLs (Table 2), which activated the metabolic pathway of bile secretion. In Figure 4, the pathway enrichment analysis indicates that the *p* value of the bile acid secretion decreased as the LYCRPL dose increased. Bile acid binding leads to fecal bile acid excretion, and bile acid synthesis is critical for the removal of cholesterol from the body because 70% of cholesterol is synthesized in vivo. Bile acids also have key roles in the regulation of postprandial lipid metabolism. The effects seem to be caused by bile acid receptors, such as farnesoid X receptor (FXR) and Takeda G protein-coupled receptor 5 (TGR5) [59]. It is speculated that LYCRPLs may activate the bile secretion by up-regulating the level of D-glucose (Figure 6E) via bile acid receptors of FXR and TGR5, thereby regulating the lipid metabolism disorder in rats.

The metabolism of cysteine is closely related to the development of hyperlipidemia. Cysteine is a precursor of protein synthesis, which is tightly regulated in the body to ensure its proper level for metabolism, while also keeping its level below the toxicity threshold [53,54,60,61]. Lysine is involved in carnitine synthesis, which is key to fatty acid metabolism. Carnitine is synthesized from lysine residues in existing proteins, transporting fatty acids to the mitochondria, where they are broken down via the TCA cycle to release energy [62,63,64,65]. Our results showed that the expression levels of palmitic acid and ethanolamine in the fecal metabolites of SD rats were significantly (*p* < 0.01) elevated after feeding with a high-fat diet. Palmitic acid is a lipid involved in fatty acid synthesis, and ethanolamine is an initial precursor for the biosynthesis of phosphatidylcholine (PC) and phosphatidylethanolamine (PE). In contrast, the ingestion of low and high doses of LYCRPLs significantly (*p* < 0.05) reduced the level of 3-methyl-2-oxovaleric acid, which is a metabolic toxin caused by the incomplete breakdown of branched-chain amino acids. Supplementation with high doses of LYCRPLs also significantly (*p* < 0.05) increased the expression of L-lysine and activated the carnitine synthesis pathway. Interestingly, treatment with low, medium, and high doses of LYCRPLs significantly (*p* < 0.05) increased the content of L-cysteine in the fecal metabolites of SD rats. Thus, we propose that L-cysteine may be a useful biomarker for the effect of LYCRPLs on rat fecal metabolites. In addition, the FoxO signaling pathway, glucagon signaling pathway, galactose metabolism, and insulin signaling pathway are also active after treatment with LYCRPLs, indicating that glucose metabolism and lipid metabolism in rats are interrelated [66,67,68]. The effect and mechanism of LYCRPLs on glucose metabolism can also be systematically investigated in the next study.

## 5. Conclusions

In summary, 101 metabolites were identified in group M. There were 54, 47, and 57 metabolites in groups GA, GB, and GC that were significantly different from that of group M, respectively. Eighteen potential biomarkers related to hyperlipidemia were screened and classified into several metabolic pathways closely related to lipid metabolism in rats. Treatment with different doses of LYCRPLs was associated with the changes in the levels of compounds, including glycolic acid, L-cysteine, glycerol 3-phosphate, 3-methyl-2-oxovaleric acid, phosphoenolpyruvic acid, L-glutamine, pantothenic acid, D-glucose, L-lysine, and oxoglutaric acid. The therapeutic targets of LYCRPLs on hyperlipidemia were mainly enriched in cysteine metabolism, carnitine synthesis, pantothenate and CoA biosynthesis, glycolysis, and bile secretion. It is speculated that LYCRPLs can regulate the lipid metabolism in SD rats via activating metabolic pathways by modulating these metabolites, and L-cysteine may be a useful biomarker for the effect of LYCRPLs on rat fecal metabolites. These results can provide theoretical support for the subsequent development of LYCRPLs as functional foods and excipients with hypolipidemic effects, as well as enrich the research system of DHA-enriched phospholipids.

## Figures and Tables

**Figure 1 foods-12-01687-f001:**
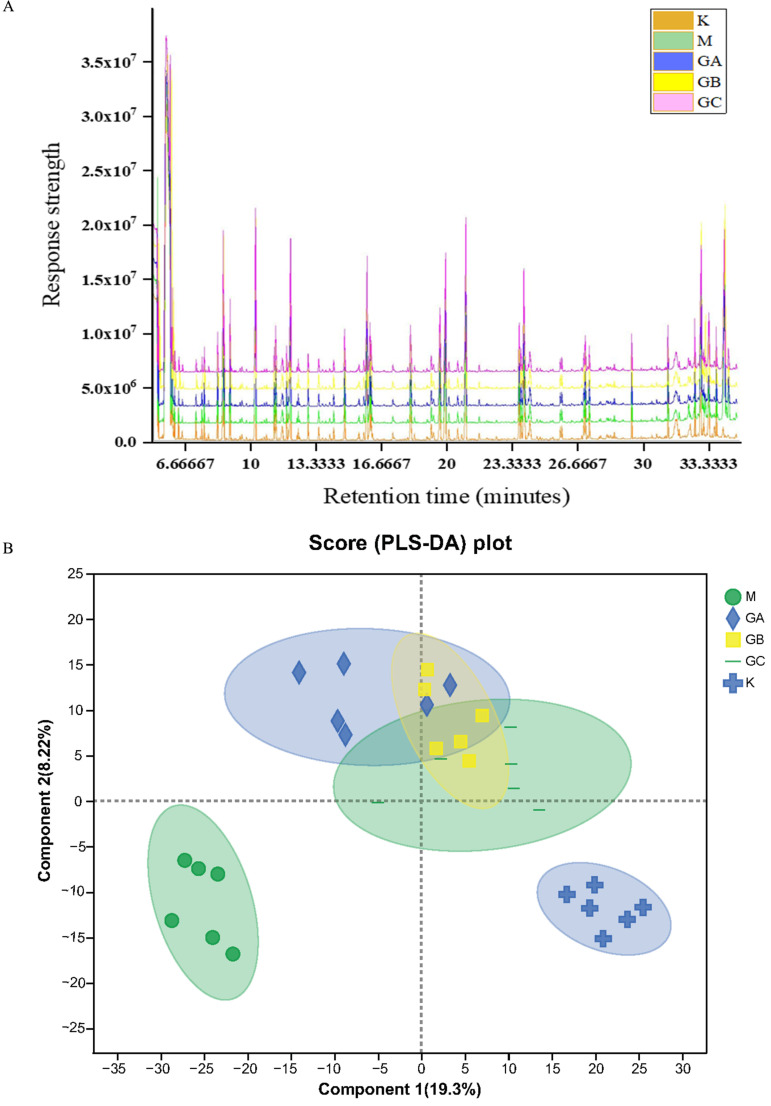
Effect of LYCRPLs on fecal metabolites in rats with a high-fat diet: (**A**) TIC overlay of QC samples; (**B**) PLS-DA results of fecal metabolites in different groups. Component 1 represents the interpretation degree of the first principal component and component 2 is the interpretation degree of the second principal component.

**Figure 2 foods-12-01687-f002:**
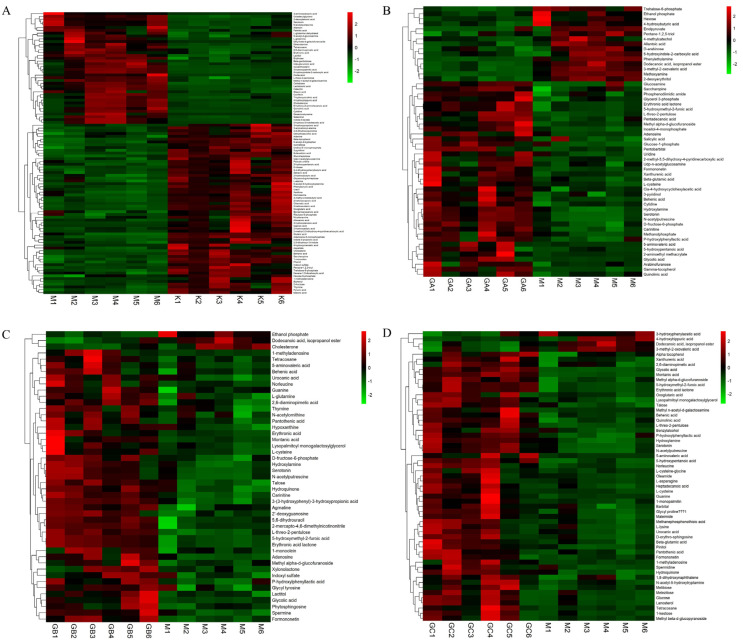
Hierarchical clustering of significantly different metabolite expression results: (**A**) clustering heatmap of samples between group K and group M; (**B**) clustering heatmap of samples between group GA and group M; (**C**) clustering heatmap of samples between group GB and group M; (**D**) clustering heatmap of samples between group GC and group M. (**A**–**D**) The colors from green to red indicates that the expression abundance levels of metabolites range from low to high. The redder the color, the higher the expression abundance of different metabolites.

**Figure 3 foods-12-01687-f003:**
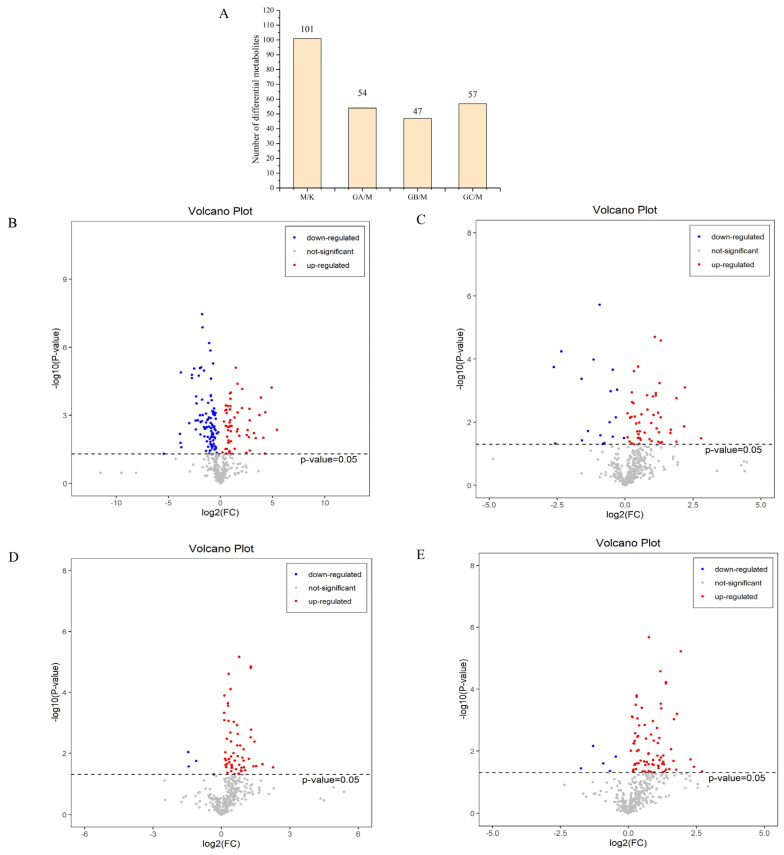
Screening of differential metabolites in the intestine of rats with a high-fat diet after treatment with LYCRPLs: (**A**) statistics showing the numbers of differential metabolites among groups; (**B**) volcano map of differential metabolites between group K and group M; (**C**) volcano map of differential metabolites between group GA and group M; (**D**) volcano map of differential metabolites between group GB and group M; (**E**) volcano map of differential metabolites between group GC and group M; (**B**–**E**) the red dots represent the significantly up-regulated differential metabolites, the blue dots represent the significantly down-regulated differential metabolites, and the gray dots represent the insignificant differential metabolites. Note: the results of the T test, *p* < 0.05 means significant.

**Figure 4 foods-12-01687-f004:**
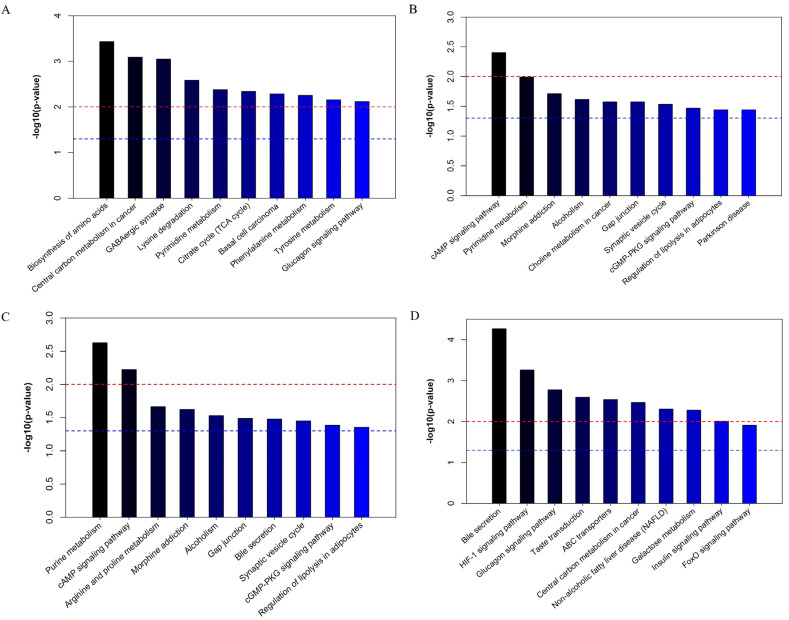
A pathway enrichment analysis of differential metabolites after treatment of LYCRPLs: (**A**) metabolic pathway enrichment map between group K and group M; (**B**) metabolic pathway enrichment map between group GA and group M; (**C**) metabolic pathway enrichment map between group GB and group M; (**D**) metabolic pathway enrichment map between group GC and group M. Note: (**A**–**D**) *p* < 0.05 means significant, and the smaller the *p* value, the more significant the difference in the metabolic pathway. The red line indicates that the *p* value is 0.01 and the blue line indicates that the *p* value is 0.05.

**Figure 5 foods-12-01687-f005:**
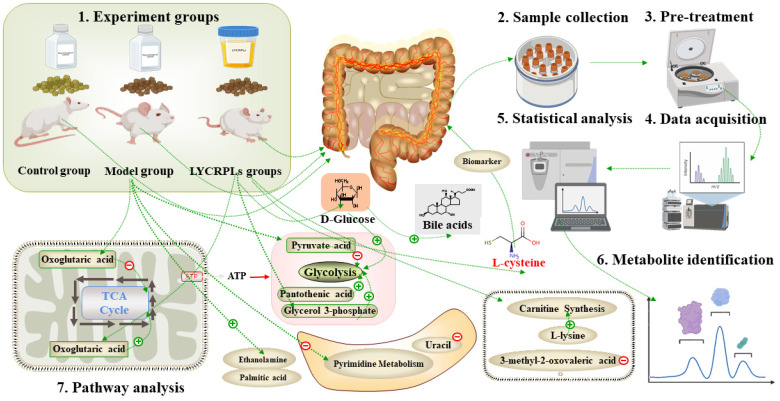
An analysis of the effect of LYCRPLs on SD rats fed with a high-fat diet based on GC/MS metabolomics.

**Figure 6 foods-12-01687-f006:**
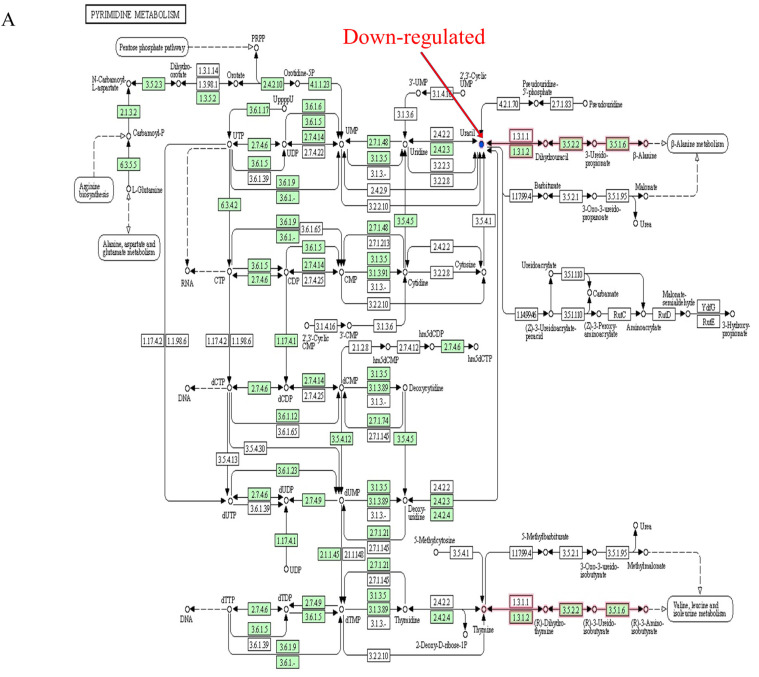
KEGG signal pathway analysis after treatment with LYCRPLs: (**A**,**B**) changes in rat fecal metabolites induced by high-fat diet in pyrimidine metabolism (**A**) and the TCA cycle (**B**); (**B**,**C**) changes in high-fat-diet rat fecal metabolites in the TCA cycle before (**B**) and after (**C**) treatment with LYCRPLs; (**D**,**E**) changes in high-fat-diet rat intestine D-glucose in terms of glycolysis (**D**) and bile secretion (**E**) after treatment with LYCRPLs. (**A**–**E**) The small circle represents metabolites, the red indicators represent up-regulation, and the blue ones represent down-regulation. The large squares represent other metabolic pathways, the small ones represent enzymes, and the small green ones represent enzymes unique to this species. The solid arrow indicates the reaction direction, and the dashed one indicates the relationship with other metabolic pathways.

**Table 1 foods-12-01687-t001:** The nutritional information for the diet.

Common Diet	Carbohydrate	Protein	Fat	Lard	Whole Egg Yolk Powder	Cholesterol	Bile Salt
	41.47%	21.06%	14.42%	0	0	0	0
High-fat diet	Carbohydrate	Protein	Fat	Lard	Whole egg yolk powder	Cholesterol	Bile salt
	32.68%	16.69%	11.03%	10%	10%	1%	0.2%

**Table 2 foods-12-01687-t002:** Differential metabolites among treatment groups.

Groups	Metabolites	Formula	Pathways	Total Score	VIP	Trend
M vs. K	Palmitic acid	C_16_H_32_O_2_	Fatty Acid Biosynthesis	99.9	1.048287271	↑ **
	Uracil	C_4_H_4_N_2_O_2_	Pyrimidine Metabolism	99.6	1.285789263	↓ **
	Ethanolamine	C_2_H_7_NO	Phospholipid Biosynthesis	98.9	1.731325782	↑ **
	Caproic acid	C_6_H_12_O_2_	Mitochondrial Beta-Oxidation of Short Chain Saturated Fatty Acids	98.6	2.317483899	↓ *
	Pyruvic acid	C_3_H_4_O_3_	Pyruvate Metabolism	96.7	2.029064881	↓ **
	L-alanine	C_3_H_7_NO_2_	Glucose-Alanine Cycle	95.8	1.150004033	↓ **
	2-hydroxybutyric acid	C_4_H_8_O_3_	Propanoate Metabolism	94.6	1.150329979	↓ **
	N-acetyl-d-glucosamine	C_8_H_15_NO_6_	Amino Sugar Metabolism	94	1.038048935	↑ *
	Oxoglutaric acid	C_5_H_6_O_5_	Citric Acid Cycle	90.8	1.872089434	↓ **
GA vs. M	Glycolic acid	C_2_H_4_O_3_	Fatty Acid Degradation	98.8	1.673615769	↑ *
	L-cysteine	C_3_H_7_NO_2_S	Cysteine Metabolism	98.8	1.123402905	↑ *
	Glycerol 3-phosphate	C_3_H_9_O_6_P	Glycolysis	95	1.109006616	↑ *
	3-methyl-2-oxovaleric acid	C_6_H_10_O_3_	Valine, Leucine, and Isoleucine Degradation	94.1	2.94484527	↓ **
	Phosphoenolpyruvic acid	C_3_H_5_O_6_P	Gluconeogenesis	90.6	1.793270632	↓ *
GB vs. M	L-cysteine	C_3_H_7_NO_2_S	Cysteine Metabolism	98.8	1.294524663	↑ *
	L-glutamine	C_5_H_10_N_2_O_3_	Glutamate Metabolism	97	1.728967115	↑ *
	Pantothenic acid	C_9_H_17_NO_5_	Pantothenate and CoA Biosynthesis	96.6	1.464407157	↑ *
GC vs. M	L-cysteine	C_3_H_7_NO_2_S	Cysteine Metabolism	98.8	1.371971343	↑ *
	D-Glucose	C_6_H_12_O_6_	Glycolysis	98.2	1.590452991	↑ *
	Pantothenic acid	C_9_H_17_NO_5_	Pantothenate and CoA Biosynthesis	96.6	1.350652592	↑ *
	L-lysine	C_6_H_14_N_2_O_2_	Carnitine Synthesis	94.3	1.249256742	↑ *
	3-methyl-2-oxovaleric acid	C_6_H_10_O_3_	Valine, Leucine, and Isoleucine Degradation	94.1	1.530958872	↓ *
	Oxoglutaric acid	C_5_H_6_O_5_	Citric Acid Cycle	90.8	1.33237846	↑ *

Note: The first column of “A vs. B” indicates the experimental group and the control group, respectively. The fifth column “total score” represents the total similarity score of this metabolite. The sixth column “VIP” is a measure of how much each metabolite contributes to the classification and identification of samples in each group, with VIP > 1 usually being used as a selection criteria for differentiating metabolites. The seventh column “trend” shows the changes in A compared to B, with ↑ indicating an increase in A compared to B and ↓ indicating a decrease. Duncan’s multiple range test was used to determine the significance of differences (* *p* < 0.05) and extremely significant differences (** *p* < 0.01) between metabolites in the different groups.

## Data Availability

The data presented in this study are available in Appendix A.

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
