# Peer review of "Effect of DHA-Enriched Phospholipids from Fish Roe on Rat Fecal Metabolites: Untargeted Metabolomic Analysis"

_foods, 2023, doi:10.3390/foods12081687_

Round 1
Reviewer 1 Report
They investigated effect of DHA-enriched phospholipids from large yellow croaker roe (LYCRPLs) on metabolites of feces using GC/MS. Comments are as follows.
1-The present study investigated the metabolites in the feces, not in the intestine. The authors need to mention that way.
2-Fig. 1: OPLS-DA is a useful analytical method to determine differences in metabolites among groups. Therefore, OPLS-DA should be performed for all groups (5 groups), not between 2 groups.
3-Bile acid secretion: The authors have interesting results with bile acid excretion, but don't seem to realize it. This reviewer reckon that bile acid secretion could be a key function of LYCRPLs. In Fig. 4, pathway enrichment analysis indicated that the p value of bile acid secretion decreased as LYCRPL dose increased. Bile acid-binding capacity leads fecal bile acid excretion and bile acid synthesis is critical for the removal of cholesterol from the body because 70% cholesterol is synthesized in vivo. Bile acids also have the key roles in regulation of postprandial lipid metabolism. These effects appear to be mediated via bile acid receptors, farnesoid X receptor (FXR) and Takeda G protein-coupled receptor 5 (TGR5). The authors need to reconsider these points to improve their paper.
Author Response
Dear Ms. Dumom Su and Reviewers,
On behalf of my co-authors, we must thank you for the critical feedback, and we appreciate editor and reviewers very much for your positive and constructive comments and suggestions on our manuscript entitled “Analysis of the effect on intestine metabolites in rats by phospholipids from fish roe based on untargeted metabolomic”. Those comments are all valuable and very helpful for revising and improving our paper, as well as the important guiding significance to our researches.
We have studied comments carefully and have made corrections which marked in the paper. We have tried our best to revise our manuscript according to the comments. Attached please find the revised version, which we would like to submit for kind consideration.
We would like to express our great appreciation to you and reviewers for comments on our manuscript.
Looking forward to hearing from you soon. Thank you and best wishes.
Yours sincerely,
Peng Liang
E-mail: liangpeng137@sina.com
Response to Reviewer #1:
Reviewer #1: They investigated effect of DHA-enriched phospholipids from large yellow croaker roe (LYCRPLs) on metabolites of feces using GC/MS. Comments are as follows.
- The present study investigated the metabolites in the feces, not in the intestine. The authors need to mention that way.
Response: Thank you very much for your constructive suggestions, we have revised the expression throughout the text to make this manuscript more scientific.
- Fig. 1: OPLS-DA is a useful analytical method to determine differences in metabolites among groups. Therefore, OPLS-DA should be performed for all groups (5 groups), not between 2 groups.
Response: Our results were analyzed with the help of the LUG database, which could only perform OPLS-DA analysis between two groups. Thank you for your suggestion, which provides a good help for us to use the OPLS-DA function more scientifically in the future.
- Bile acid secretion: The authors have interesting results with bile acid excretion, but don't seem to realize it. This reviewer reckon that bile acid secretion could be a key function of LYCRPLs. In Fig. 4, pathway enrichment analysis indicated that the p value of bile acid secretion decreased as LYCRPL dose increased. Bile acid-binding capacity leads fecal bile acid excretion and bile acid synthesis is critical for the removal of cholesterol from the body because 70% cholesterol is synthesized in vivo. Bile acids also have the key roles in regulation of postprandial lipid metabolism. These effects appear to be mediated via bile acid receptors, farnesoid X receptor (FXR) and Takeda G protein-coupled receptor 5 (TGR5). The authors need to reconsider these points to improve their paper.
Response: We have revised this part carefully as the reviewer suggested. Thank you very much for your suggestions on the bile acids section, which have greatly helped us improve our discussion. Thank you again for your constructive and valuable comments!

Reviewer 2 Report
L34-37: Seems to be missing something because the meaning of the sentence is not complete.
L 10: Reference (s)?
L 78: Please give full name
The introduction regarding the elements contained in the experimental protocol is considered incomplete, containing generalities.
L 108: “After the experimental SD rats adapted to the environment”. Before or after? Also please explain what is SD, SPF.
How the LYCRPLs was given to the SD rats?
In this particular experiment can someone extract metabolomics data only by looking at biomolecules in the feces without looking at the microflora and any metabolites produced by its microorganisms?
L 133: References?
The resolution of Figures 2,3 & 6, is very poor and one cannot study the data presented.
The discussion, as well as the introduction, contains generalities without presenting possible mechanisms to interpret the data captured in the present study.
Author Response
Dear Ms. Dumom Su and Reviewers,
On behalf of my co-authors, we must thank you for the critical feedback, and we appreciate editor and reviewers very much for your positive and constructive comments and suggestions on our manuscript entitled “Analysis of the effect on intestine metabolites in rats by phospholipids from fish roe based on untargeted metabolomic”. Those comments are all valuable and very helpful for revising and improving our paper, as well as the important guiding significance to our researches.
We have studied comments carefully and have made corrections which marked in the paper. We have tried our best to revise our manuscript according to the comments. Attached please find the revised version, which we would like to submit for kind consideration.
We would like to express our great appreciation to you and reviewers for comments on our manuscript.
Looking forward to hearing from you soon. Thank you and best wishes.
Yours sincerely,
Peng Liang
E-mail: liangpeng137@sina.com
Response to Reviewer #2:
Reviewer #2: Comments and Suggestions for Authors
- L34-37: Seems to be missing something because the meaning of the sentence is not complete.
Response: We have revised the sentence to make it clearer as the reviewer suggested. Thank you so much!
- L 10: Reference (s)?
Response: Sorry, I can't find the location you are pointing to. L 10 corresponds to the address of the corresponding author.
- L 78: Please give full name.
Response: The full name has been added for QC. Thank you very much for your carefulness!
- The introduction regarding the elements contained in the experimental protocol is considered incomplete, containing generalities.
Response: We have revised this part to make the introduction clearer as reviewer suggested.
- L 108: “After the experimental SD rats adapted to the environment”. Before or after? Also please explain what is SD, SPF.
Response: Thank you for your question! After the experimental Sprague Dawley (SD) rats adapted to the environment, 50 male SD rats (Specific pathogen free, SPF, 120 ± 10g) were randomly divided into five groups in our manuscript. After the experimental rats are purchased, they may not adapt to the unfamiliar environment, which will affect the weight gain of the experimental rats. Therefore, we chose to group the experimental rats after they had acclimatized to the environment, which would minimize the differences between groups of rats at the beginning of the experiment. And the full name has been added for SD and SPF.
- How the LYCRPLs was given to the SD rats?
Response: The samples were given by intragastric administration (dose: 0.01 mL·g-1) at a fixed time every day for 8 weeks. This section has also been added at the corresponding place in the manuscript. Thank you for your professional advice!
- In this particular experiment can someone extract metabolomics data only by looking at biomolecules in the feces without looking at the microflora and any metabolites produced by its microorganisms?
Response: Thank you very much for your question! In our previous study [1], we found that DHA-enriched phospholipids from large yellow croaker roe regulated lipid metabolic disorders and gut microbiota imbalance in SD rats with a high-fat diet. Feces are the main substance excreted in the gut, so we collected fresh feces and analyzed the differential metabolites between groups in this study, hoping to better explain the effect of DHA-enriched phospholipids from large yellow croaker roe on lipid regulation in rats. Silke Matysik et al. reviewed the related research on Metabolomics of fecal samples, illustrating the feasibility of fecal metabolomics research. Fecal samples, that contain unabsorbed metabolites, offer a direct access to the outcome of diet – gut microbiota metabolic interactions. Hence, they are a useful addition to measure the ensemble of endogenous and microbial metabolites, also referred to as the hyperbolome [2]. The relevant references are as follows.
[1] Xiaodan Lu et al. DHA-enriched phospholipids from large yellow croaker roe regulate lipid metabolic disorders and gut microbiota imbalance in SD rats with a high-fat diet. Food & Function. 12.11 (2021): 4825-4841.
[2] Silke Matysik et al. Metabolomics of fecal samples: a practical consideration. Trends in Food Science & Technology. 57 (2016): 244-255.
- L 133: References?
Response: References have been added where appropriate. Thank you!
- The resolution of Figures 2,3 & 6, is very poor and one cannot study the data presented.
Response: We have provided higher resolution images for Figures 2, 3 and 6 in the attached file as suggested by the reviewer.
- The discussion, as well as the introduction, contains generalities without presenting possible mechanisms to interpret the data captured in the present study.
Response: We have revised the discussion and introduction in a more detailed form as suggested. Thank you again for your valuable time on our manuscript!

Reviewer 3 Report
Very interesting and novel study. The methodology used is consistent with the purpose of the study. The experiments carried out are sufficient. The results support the discussion. However, I have the following comments.
I. Major Comments.
1. The manuscript is well written. But it was difficult for me to review the figures. The size of the letters is very small. This must be corrected by the authors.
2. In the introduction I suggest briefly including, among the benefits of n-3 PUFAs, their effects on the brain (nervous system) and the liver. PMID: 30968772; PMID: 33803760 .
3. Table 1. Increase the size of the letters.
4. In the discussion it is necessary to include aspects related to the effects of lipids on the intestinal microbiota, and then the specific effects of n-3 PUFAs. In addition, I suggest including some mechanisms related to the inflammatory response and oxidative stress in the intestine.
5. Would an increase in the absorption of n-3 PUFAs be expected, considering that phospholipids were used? discuss this point.
6. What application could have these results?
II. Minor comments.
1. Improve the wording of the objective of the study.
2. Include the nutritional information of the diet used (energy as proteins, fat and carbohydrates). So, like the intervention time with the diet.
3. Figure 5 is very good. But, I suggest increasing the size of the letters.
Author Response
Dear Ms. Dumom Su and Reviewers,
On behalf of my co-authors, we must thank you for the critical feedback, and we appreciate editor and reviewers very much for your positive and constructive comments and suggestions on our manuscript entitled “Analysis of the effect on intestine metabolites in rats by phospholipids from fish roe based on untargeted metabolomic”. Those comments are all valuable and very helpful for revising and improving our paper, as well as the important guiding significance to our researches.
We have studied comments carefully and have made corrections which marked in the paper. We have tried our best to revise our manuscript according to the comments. Attached please find the revised version, which we would like to submit for kind consideration.
We would like to express our great appreciation to you and reviewers for comments on our manuscript.
Looking forward to hearing from you soon. Thank you and best wishes.
Yours sincerely,
Peng Liang
E-mail: liangpeng137@sina.com
Response to Reviewer #3:
Reviewer #3: Very interesting and novel study. The methodology used is consistent with the purpose of the study. The experiments carried out are sufficient. The results support the discussion. However, I have the following comments.
- Major Comments.
- The manuscript is well written. But it was difficult for me to review the figures. The size of the letters is very small. This must be corrected by the authors.
Response: Thank you very much for your suggestion! We have provided higher resolution figures in the attached file for better review as suggested.
- In the introduction I suggest briefly including, among the benefits of n-3 PUFAs, their effects on the brain (nervous system) and the liver. PMID: 30968772; PMID: 33803760.
Response: Thank you very much for your constructive comments! The benefits of n-3 PUFAs have been added in the introduction as the reviewer suggested, which would make this manuscript more attractive. And the corresponding references have also been cited in the manuscript.
[1] Rodrigo Valenzuela et al. "Targeting n-3 polyunsaturated fatty acids in non-alcoholic fatty liver disease." Current medicinal chemistry. 27.31 (2020): 5250-5272. (PMID: 30968772)
[1] Verónica Sambra et al. "Docosahexaenoic and arachidonic acids as neuroprotective nutrients throughout the life cycle." Nutrients. 13.3 (2021): 986. (PMID: 33803760)
- Table 1. Increase the size of the letters.
Response: The size of the letters in Table 1 has been increased for better review. Thank you for your suggestion!
- In the discussion it is necessary to include aspects related to the effects of lipids on the intestinal microbiota, and then the specific effects of n-3 PUFAs. In addition, I suggest including some mechanisms related to the inflammatory response and oxidative stress in the intestine.
Response: The aspects related to the effects of lipids on the intestinal microbiota and the effects of n-3 PUFAs have been added as reviewer suggested. And the inflammatory response and oxidative stress in the intestine have also been included in the discussion section. Thank you so much for your valuable comments!
- Would an increase in the absorption of n-3 PUFAs be expected, considering that phospholipids were used? discuss this point.
Response: Thank you very much for your valuable advice. In our previous study [1], we found that DHA-enriched phospholipids from large yellow croaker roe regulated lipid metabolic disorders and gut microbiota imbalance in SD rats with a high-fat diet. Feces are the main substance excreted in the gut, so we collected fresh feces and analyzed the differential metabolites between groups in this study, hoping to better explain the effect of DHA-enriched phospholipids from large yellow croaker roe on lipid regulation in rats. Therefore, whether the absorption of n-3 PUFAs is increased was not discussed in this study. However, it is a very interesting question, which provides a new idea for our further research on LYCRPLs. In the follow-up research, we are going to design reasonable experiments to verify this conjecture. Thanks again! The relevant reference is as follows.
[1] Xiaodan Lu et al. DHA-enriched phospholipids from large yellow croaker roe regulate lipid metabolic disorders and gut microbiota imbalance in SD rats with a high-fat diet. Food & Function. 12.11 (2021): 4825-4841.
- What application could have these results?
Response: Analysis Base File Converter software was used to convert the raw data (.D format) to .abf format, and then the .abf data were imported into the MD-DIAL software for data processing. Metabolites were annotated through LUG database which is the untargeted database of GC-MS from Shanghai Luming Biological Technology co., LTD (Shanghai, China). And thank you again for your arduous work and all the instructive advices.
- Minor comments.
- Improve the wording of the objective of the study.
Response: We have revised the wording of the objective of the study as reviewer suggested. Thank you for your suggestion!
- Include the nutritional information of the diet used (energy as proteins, fat and carbohydrates). So, like the intervention time with the diet.
Response: The nutritional information of the diet used in this manuscript has been included in Table 1 as suggested. The samples were given by intragastric administration (dose: 0.01 mL·g-1) at a fixed time every day for 8 weeks, and the intervention time has been added in Animal Treatment and Sample Collection section.
- Figure 5 is very good. But, I suggest increasing the size of the letters.
Response: The letter size in Figure 5 has been increased as suggested to make it easier to read. And other unclear figures have also been replaced. Thank you again for your arduous work and all the instructive advices!

Round 2
Reviewer 1 Report
I cannot understand why the authors revised Figure 1 according to my comment. It is not acceptable. They should do now, not in the future.
My privious comment: Fig 1: OPLS-DA is a useful analytical method to determine differences in metabolites among groups. Therefore, OPLS-DA should be performed for all groups (5 groups), not between 2 groups.
Author Response
Dear Ms. Dumom Su and Reviewers,
On behalf of my co-authors, we must thank you for the critical feedback, and we appreciate editor and reviewers very much for your positive and constructive comments and suggestions on our manuscript entitled “Analysis of the effect on intestine metabolites in rats by phospholipids from fish roe based on untargeted metabolomic”. Those comments are all valuable and very helpful for revising and improving our paper, as well as the important guiding significance to our researches.
We have studied comments carefully and have made corrections which marked in the paper. We have tried our best to revise our manuscript according to the comments. Attached please find the revised version, which we would like to submit for kind consideration.
We would like to express our great appreciation to you and reviewers for comments on our manuscript.
Looking forward to hearing from you soon. Thank you and best wishes.
Yours sincerely,
Peng Liang
E-mail: liangpeng137@sina.com
Response to Reviewer #1:
Reviewer #1: I cannot understand why the authors revised Figure 1 according to my comment. It is not acceptable. They should do now, not in the future.
My privious comment: Fig 1: OPLS-DA is a useful analytical method to determine differences in metabolites among groups. Therefore, OPLS-DA should be performed for all groups (5 groups), not between 2 groups.
Response: Fig. 1 has been modified as suggested and corresponding changes have been made in the manuscript. As we stated before, the results were analyzed with the help of the LUG database, which could only perform OPLS-DA analysis between two groups. Therefore, the PLS-DA analysis was performed for all groups to determine differences in metabolites among groups in this study, since OPLS-DA is a method improved on the basis of PLS-DA. Thank you very much for your valuable suggestions and rigorous scientific attitude! Sincerely hope that our modification can meet your requirements.

Reviewer 2 Report
The authors have responded satisfactorily to all my comments. But, one final touch, following your answer;" Hence, they are a useful addition to measure the ensemble of endogenous and microbial metabolites, also referred to as the hyperbolome", please discuss the article's methodological limitations and planned future aspects.
And also;
- L 40: Reference (s)?
Thank you
Author Response
Dear Ms. Dumom Su and Reviewers,
On behalf of my co-authors, we must thank you for the critical feedback, and we appreciate editor and reviewers very much for your positive and constructive comments and suggestions on our manuscript entitled “Analysis of the effect on intestine metabolites in rats by phospholipids from fish roe based on untargeted metabolomic”. Those comments are all valuable and very helpful for revising and improving our paper, as well as the important guiding significance to our researches.
We have studied comments carefully and have made corrections which marked in the paper. We have tried our best to revise our manuscript according to the comments. Attached please find the revised version, which we would like to submit for kind consideration.
We would like to express our great appreciation to you and reviewers for comments on our manuscript.
Looking forward to hearing from you soon. Thank you and best wishes.
Yours sincerely,
Peng Liang
E-mail: liangpeng137@sina.com
Response to Reviewer #2:
Reviewer #2: The authors have responded satisfactorily to all my comments. But, one final touch, following your answer;" Hence, they are a useful addition to measure the ensemble of endogenous and microbial metabolites, also referred to as the hyperbolome", please discuss the article's methodological limitations and planned future aspects.
Response: In this article, metabolomics data were obtained solely by looking at biomolecules in feces. Although it is a useful addition to our previous research on the gut microbiome, there are still limitations. Therefore, we are going to design reasonable experiments to verify the effect of DHA-enriched phospholipids from large yellow croaker roe (LYCRPLs) on endogenous metabolites in the follow-up researches. And the results of this study will be further integrated with that of endogenous and microbial metabolites to better reveal the effect mechanism of LYCRPLs in regulating lipid metabolism. Thank you very much for your constructive suggestions, which provide a new idea for our further researches on LYCRPLs!
- L 40: References?
Response: References have been added where appropriate. Thank you again for your valuable time on our manuscript!

Reviewer 3 Report
Authors answered all my comments.
Author Response
Dear Ms. Dumom Su and Reviewers,
On behalf of my co-authors, we must thank you for the critical feedback, and we appreciate editor and reviewers very much for your positive and constructive comments and suggestions on our manuscript entitled “Analysis of the effect on intestine metabolites in rats by phospholipids from fish roe based on untargeted metabolomic”. Those comments are all valuable and very helpful for revising and improving our paper, as well as the important guiding significance to our researches.
We have studied comments carefully and have made corrections which marked in the paper. We have tried our best to revise our manuscript according to the comments. Attached please find the revised version, which we would like to submit for kind consideration.
We would like to express our great appreciation to you and reviewers for comments on our manuscript.
Looking forward to hearing from you soon. Thank you and best wishes.
Yours sincerely,
Peng Liang
E-mail: liangpeng137@sina.com
Response to Reviewer #3:
Reviewer #3: Authors answered all my comments.
Response: Thank you again for your valuable time on our manuscript!
